# Does individual advocacy work?: A research and evaluation protocol for a youth anti-sex trafficking program

**Mary K. Twis****¹\*, Andrea Cimino², Morgan Files³**

**1** Department of Social Work, Texas Christian University, Fort Worth, Texas, United States of America,
**2** Danger Assessment Training and Technical Assistance Center, Johns Hopkins University, Baltimore, Maryland, United States of America, **3** Traffick911, Addison, Texas, United States of America

\* Mary.twis@tcu.edu

## Abstract

### Introduction

Thousands of youth are sexually trafficked each year worldwide. Increased public attention to the commercial sexual exploitation (CSE) of children has resulted in the rapid deployment of hybrid community public health and social service programs for these vulnerable youth. Research on the effectiveness of these advocacy programs is lacking, particularly whether they decrease psychosocial distress and increase readiness to leave CSE.

### Methods and analysis

Cisgender girls under age 18 at the time of CSE, and who were identified as at-risk for sex trafficking revictimization, were included in an evaluation of an anti-trafficking advocacy program in the North Texas region of the United States. The program includes crisis response, case management, referral, and mentoring services in collaboration with multi-disciplinary team (MDT) responses to identified youth sex trafficking. Case management notes, needs assessments and individualized treatment plans were collected at intake and every 30 days until study conclusion. Standardized surveys, including the Multidimensional Scale of Perceived Social Support (MSPSS), the Coping Self-Efficacy Scale, and the University of Rhode Island Change Assessment (URICA) were collected at intake and every 180 days until the study concluded. Analyses included descriptive statistics, paired t-tests, chi-square, multivariate linear and logistic regressions, Poisson regressions, and latent profile analysis.

### Ethics and dissemination

This study was approved by the Texas Christian University's Institutional Review Board (IRB). Results of this study will be presented to the scientific community at conferences and in peer-reviewed journals and non-scholarly outlets such as public health and social service conferences.

**Data Availability Statement:** No datasets were generated or analyzed during the current study. All relevant data from this study will be made available upon study completion.

**Funding:** This project was supported by a Victims of Crime Act (VOCA) federal pass-through grant awarded to the Office of the Texas Governor (OOG) Child Sex Trafficking Team and Traffick911. The sponsor of this protocol – Traffick911 – has assisted with the study design and data collection plan described in this protocol. Traffick911 administrators have approved of the protocol described herein. The original funders of this project, including VOCA and the OOG, had no role in study design, data collection and analysis, decision to publish, or preparation of the manuscript. The opinions, findings, conclusions, and recommendations expressed in this publication are those of the authors and do not necessarily reflect the views of VOCA, OOG, or Traffick911. (www.traffick911.com).

**Competing interests:** The authors have declared that no competing interests exist.

# Introduction

Sex trafficking is a severe human rights violation that disproportionately affects vulnerable children and youth. It is internationally defined as the recruitment, transportation, transfer, harboring or receipt of persons by threat, force, coercion, abduction, fraud, deception, and abuse of power for commercial sexual exploitation (CSE) [1]. In the United States, sex trafficking is similarly defined in the Trafficking Victims Protection Act [2] and its reauthorizations [3–7]. The TVPA [2] articulates that any individual under the age of 18 who is induced to perform a sex act in exchange for money, goods, or services, even without the use of force, fraud, or coercion, is considered a victim of sex trafficking and CSE. CSE victims can be exploited in on-street prostitution, brothels, sex tourism, mail-order-bride trade and early marriage, pornography, cyber enticement, stripping, and performing in sexual venues such as peep shows or clubs [8]. It is not possible to know how many persons are sexually trafficked throughout the world [9], but estimates from the United States government suggest that thousands of youth are sex trafficked each year, based upon the prevalence of known CSE risk factors like child abuse/neglect and child welfare involvement, juvenile justice involvement, youth homelessness, and running away [10].

Research has well documented the poor outcomes of children who are commercially sexually exploited. These include sexually transmitted infections (STIs) including HIV, pregnancy, malnutrition, and untreated chronic medical conditions, as well as long-lasting mental health effects including posttraumatic stress disorder (PTSD), depression, suicide, anxiety, and alcohol and drug addiction [11–15]. Research indicates that negative sequalae can be severe and persistent the younger and longer a person is commercially sexually exploited [16–20].

Given the evidence suggesting age-dependent outcomes, programs that focus on early secondary intervention with this population are particularly important. Front line workers including health care providers, law enforcement, and child protective services (CPS) play an important role in identifying sex trafficking victims and assisting them to leave CSE through appropriate referrals to social and healthcare services. This article describes the study protocol for the evaluation of a community-based anti-sex trafficking intervention program for cisgender girls. The program, run by an anti-sex trafficking agency called Traffick911, functions on the assumption that personalized advocacy, crisis response, and case management increases the likelihood that sex trafficked youth can successfully exit commercial sexual exploitation and increase their resilience towards potential revictimization. The goals of the study were to explore whether sex trafficked cisgender girls who receive individualized advocacy services experience changes in social support, self-efficacy, and readiness to change the behaviors that predict sex trafficking revictimization. The Intentions to Exit Prostitution (IEP) model [21–23] suggests that these types of changes promote an eventual exit from sex trafficking. Although the IEP model was originally developed for adult street-based sex workers, it has also been adapted and tested with commercially sexually exploited youth [24].

# Methods

## Ethics

All study procedures were approved by Texas Christian University's IRB, and by Traffick911's internal review processes. Since all data were de-identified prior to transfer to the research team, the study is considered non-human subjects research.

## The intervention program

Traffick911 provides advocacy services to hundreds of North Texas cisgender girls identified as sex trafficking victims and/or at high-risk for sex trafficking revictimization due to running

away, repeated sexual abuse and exploitation history, and/or the Commercial Sexual Exploitation–Identification Tool (CSE-IT). It was founded in 2009 to "free youth from sex trafficking through trust-based relationships" [25] and it receives funding for its services from the Office of the Texas Governor's Child Sex Trafficking Team (CST), the Victims of Crime Act, and private donors. In 2016, the CST's focus was on creating Multi-Disciplinary Teams (MDTs) that had the capacity to implement coordinated, evidence-based, and trauma-informed sex trafficking survivor services across multiple systems, including law enforcement, CPS, healthcare systems, and other related services [26]. As a result, Traffick911 is one of the state's primary regional service providers, as it plays a central role on several different MDTs in its four-county service area. Although specific roles and responsibilities vary by MDT, Traffick911 helps coordinate services provided by MDT partners like CPS, juvenile justice, law enforcement, child advocacy centers, and more. The agency's role on these MDTs is typically to advocate for clients' needs and ensure they do not fall through gaps in service delivery systems. Traffick911 further ensures that there are trained advocates who can respond to community, child welfare, or law enforcement reports of child sex trafficking within the North Texas region. The agency urgently responds to positive child sex trafficking identification by sending advocates to the location of the child or youth within 90 minutes of referral.

Traffick911 advocacy interventions consist of three lines of effort: (1) client direct services, (2) community partnerships, and (3) organizational capacity. The short- and long-term target outcomes for each effort are listed in the Program Logic Model in Fig 1. The research team

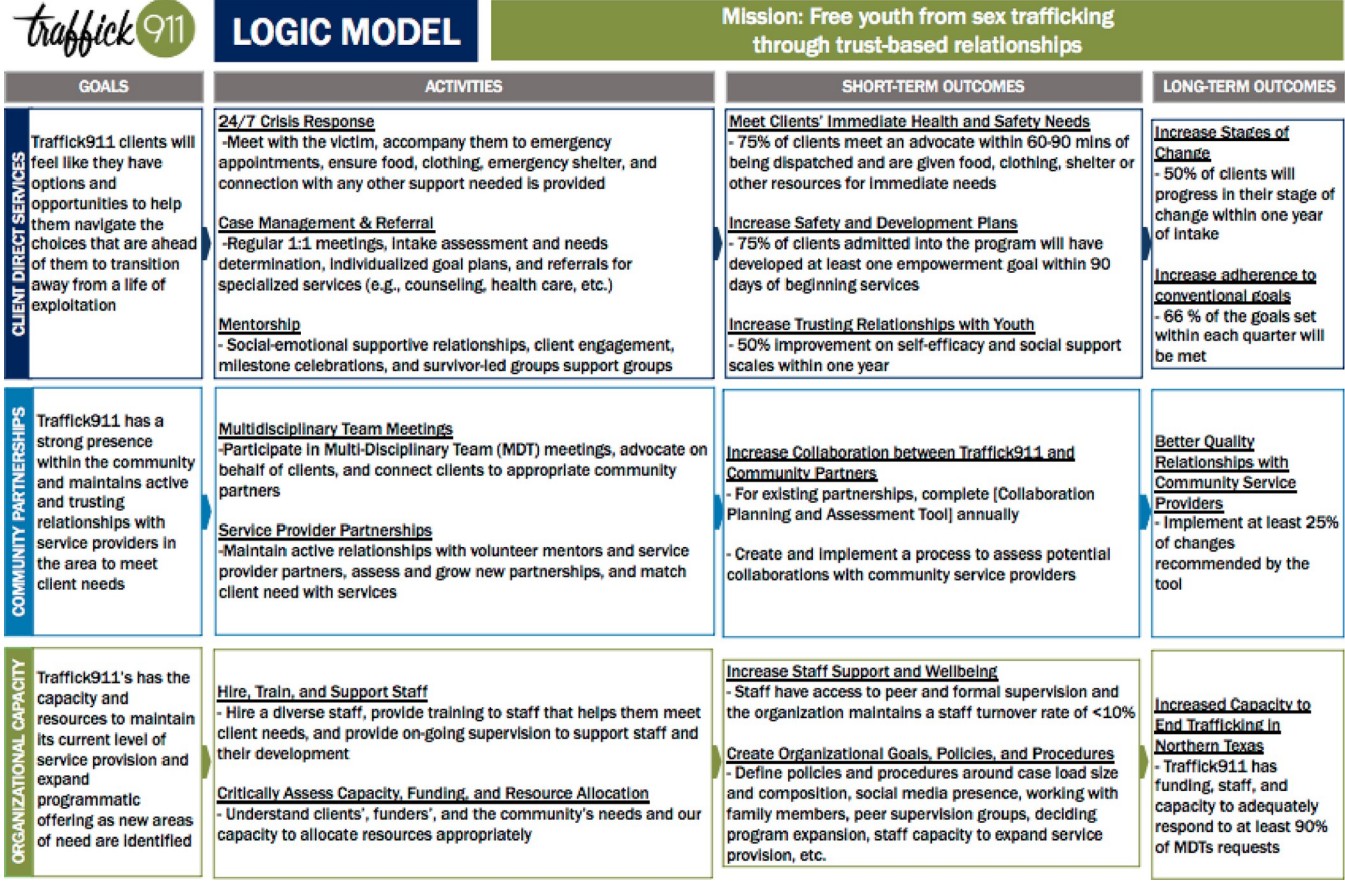

**Fig 1. Program logic model.** Traffick911 logic model demonstrating agency's theory of change.

collaboratively developed the logic model with Traffick911 immediately prior to the launch of the evaluation described herein, as it was meant to both a) capture the characteristics of the agency's central Voice & Choice Advocacy Program for easy-to-understand dissemination to other service providers, and b) guide all agency evaluation efforts. Since the Voice & Choice Advocacy Program is the central offering of Traffick911, the logic model can be understood as both a programmatic logic model and a summary of the agency's overarching theory of change.

The client direct service goal is for clients to transition away from the life of exploitation and to feel that they have options and opportunities to help them navigate the choices that are ahead of them. Activities include 24/7 crisis response services (e.g., meet clients in the community, provide food, clothing, emergency shelter, and other essentials), advocacy and case management, referral services, and mentorship to build relationships and provide socio-emotional support within survivors' own communities. Although a presentation of evaluation results falls outside the scope of this protocol, early analyses show that of the 95 clients included in the evaluation, each client had received roughly 93 advocacy services during their time in the program. The community partnership line of effort entails building and maintaining active and trusting relationships with service providers in the area to meet client needs. Activities include participating in MDT meetings and maintaining and growing volunteer mentors with local service provider partners. The goal of the final line of effort, organizational capacity, is to maintain the program's current level of service provision and expand programmatic offerings as new areas of need are identified. Activities include hiring, training, and supporting program staff and strategic planning regarding the organization's capacity, funding, and resourcing abilities.

## Study design

An outcome evaluation plan was developed by the research team in collaboration with the agency. Traffick911 was selected for this outcome evaluation because it received grant funding from the CST to explore its services and to potentially develop a service and evaluation model for other grantees to follow. Traffick911 received this grant because of its capacity to manage an evaluation, its location within a highly-populated region of Texas, its history of providing CSE advocacy services, and its positive reputation with the CST and area MDTs. The evaluation focused on measuring the anti-sex trafficking advocacy program's impact on participants' behaviors and mental health outcomes prior to and while receiving individualized advocacy services. The study design was one-group, quasi-experimental double pre/posttest design wherein program participants wee surveyed by program staff at three timepoints: baseline, 6-months, and 12-months. Additionally, process metrics such as case note counts that track client contacts/service provision (a measure of dosage) and completion of individual development plans were collected at least monthly. The study period ran from November 2020 to November 2021. Although the study was conducted in the midst of the COVID-19 pandemic, all activities described in this protocol are essential for future evaluation efforts–even when the pandemic eventually recedes. Due to the pandemic, some of the data collected for this study were collected via phone or Zoom. It would be ideal in future studies for all data to be collected in-person.

## Procedures

All collected data were input by program staff into two separate data management systems. Survey data were collected in Qualtrics, and demographic and case note data were collected in Apricot, an online database software for human service organizations. Program staff then ran reports in each of these data management systems to produce Excel files containing large

amounts of de-identified client data. All files were linked into a master database by utilizing unique client IDs across all reports. Data were only transferred to research team personnel in the form of Excel files once the data were completely de-identified. Since all data were de-identified prior to transfer, the study was considered non-human subjects research by Texas Christian University's institutional review board (IRB), and the study was thus exempt from IRB oversight (IRB determination for study 1920–279 provided on August 24, 2020). Once all data were fully transferred to investigators in November 2021, the research team conducted secondary data analysis. Analysis began in December 2021 and remains ongoing until April 2022.

## Study site

Four counties in North Texas comprised the study site, including Dallas, Tarrant, Collin, and Denton counties. These counties include Dallas and Fort Worth, which are two of the largest cities in the state of Texas and the United States. Dallas, Tarrant, Collin, and Denton counties also include numerous small, medium, and large cities, which together comprise what is locally known as the DFW Metroplex. The Metroplex is one of the fastest-growing areas in the United States, with a combined population of over 7,000,000 people [27]. Estimates suggest that roughly one out of every four Texans lives in the North Texas region, with most of these individuals living in Dallas and Tarrant counties [27]. Importantly for this protocol, over 40% of the population in Dallas county identifies as Hispanic or Latinx, and 36.8% of Dallas residents are native Spanish speakers [28].

While it is difficult to accurately estimate the number of youth who have been sexually trafficked in this region, as well as the number of these youth who are at-risk for revictimization, one widely-circulated study conducted by University of Texas researchers found that as many as 79,000 Texans under the age of 25 have been sex trafficked at some point in their lives [29]. Given the large population of the region, it is likely that many of these estimated CSE victims live in Dallas, Tarrant, Collin, and Denton counties. However, the number of reported sex trafficked victims in Texas is actually much lower than this estimate, with the Texas Department of Family and Protective Services [30] reporting only 697 allegations of sex trafficking in 2017. The ongoing discrepancies between estimates and reports continue to demonstrate how difficult it is to accurately assess both global and national human trafficking rates, as well as sex trafficking rates at local, city, or regional levels. Estimate research is difficult to conduct because of the hidden nature of sex trafficking and commercial sexual exploitation, as well as the ethical dilemmas that arise when undertaking research with child sex trafficking survivors [31, 32]. Substantiated reports do suggest, however, that the problem exists in North Texas.

Traffick911 has one office in its four-county service area, from which it manages and deploys its services across the region. Mobile crisis units travel to meet with clients when referrals are made to Traffick911 by MDT partners in each of the four counties. Program staff also make significant efforts to meet with clients in their own communities, rather than at a central location. As a result, data were collected at various locations throughout Traffick911's four-county service area, typically within identified survivors' communities.

## Participant inclusion criteria

Clients were considered for study inclusion if they were actively receiving Traffick911 services during the evaluation time period. When data collection was complete, a total of 95 clients were included in the sample, which was within the 80–120 client sample size goal the research team initially set for the study. Although Traffick911 completed intakes with more than 95 clients during the study period, some clients dropped out of the sample because they did not continue to receive services post-intake.

Individuals are eligible for Traffick911 services if they are a) identified as sex trafficking survivors under the age of 18 at time of CSE, or b) identified as youth at very high risk for sex trafficking or sex trafficking revictimization before the age of 18, due to assessments of runaway behavior and/or repeated sexual abuse and exploitation history. Often, youth are considered "at-risk" for trafficking only to later be confirmed as trafficking survivors when they are more willing to discuss their experiences with advocates or law enforcement. The CSE-IT tool, which is used to assess trafficking risk in Texas, tabulates a score of "clear concern" when a youth is identified as high-risk for sex trafficking. The CSE-IT tool has been validated, such that "clear concern" is predictive of a later confirmation of trafficking [33]. For this reason, high-risk youth who are identified as such by the CSE-IT are provided services and evaluation measures just the same as confirmed trafficking survivors.

In order to qualify for services, program participants who are under the age of 18 must have a parent or caregiver provide consent for services. Notably, requiring parental consent for services might have prevented familial trafficking victims from receiving services or participating in the evaluation. Identified victims can continue to receive Traffick911 services after reaching the age of 18. Additionally, Traffick911 services are targeted towards cisgender girls at this time. As the agency explores how to provide outreach to cisgender boys and trans youth, there were only two cisgender boys and one trans youth on agency caseloads at the time of data collection. For the purposes of this study, these cisgender boys and trans clients were excluded from the sample, since between-group variations can complicate secondary data analysis. It should be noted, however, that future applications of this protocol should attempt to include cisgender boys and trans youth in data collection and analysis, as the exclusion of these subpopulations limits the scope of study findings and widens the systemic research gaps between cisgender girls, cisgender boys, and trans youth within the CSE field.

## Study instruments

All study instruments were internally administered measures meant to gather outcome information related to changes in clients' key outcomes and progress towards reaching individual goals and obtaining basic needs. Study instruments were available in both English and Spanish. Although all Spanish-language surveys in this protocol were checked and validated by bilingual practitioners working in the field of gender-based violence, Spanish-language surveys ought to be completed with the help of professionals to ensure survivors understand the items correctly, as some nuanced words related to sex trafficking might be lost or confused in translation. All data were collected and de-identified by program staff and are currently under analysis by the research team as secondary data. See Table 1 below for a snapshot of all outcome evaluation data sources, as well as an overview of when these data were collected and where the data were managed.

**Table 1. Measures administered to clients by time point.**

| Variable | Measure | Data Management | Baseline | Every Month | Six Months | Twelve Months |
|---|---|---|---|---|---|---|
| Demographics | Intake Forms | Apricot | x | | | |
| Contacts and services | Case Note Forms | Apricot | | x | | |
| Individual development plan | Goals Form | Apricot | | x | | |
| Readiness to change | URICA | Qualtrics | x | | x | x |
| Social support | MSPSS | Qualtrics | x | | x | x |
| Self-efficacy | Coping-Self Efficacy Scale | Qualtrics | x | | x | x |

Measures administered by time frame and data management system used.

**Demographics.** Program staff gathered standard demographic characteristics from clients, including age, gender, race/ethnicity, and education, and then they input this data into an agency case management database called Apricot. Staff also gathered additional information about clients' sexual orientation and secondary victimization history (i.e., child abuse, sexual assault). This data was obtained from the client intake forms.

**Case notes.** Program staff also collected additional information on clients' interactions with Traffick911 staff and the services clients received. These data were collected through a case note form in the agency's online data management system. See S1 Fig for an illustration of the information that was gathered through the case note form. This standardized approach to gathering information related to client interactions and client services allowed Traffick911 staff to capture this data quantitatively. The case note form also has a narrative field for qualitative data related to clients' interactions with staff and agency services.

The field labeled "type of contact" includes the following options: (a) attempt to contact, (b) communication with caregiver, (c) communication with client, (d) communication with MDT partner, (e) communication with outside service provider, (f) crisis response, (g) joint communication, and (h) non-crisis response. The "type of contact" field is currently used in the secondary data analysis to quantify clients' doses of the advocacy program. The "type of update" field was utilized internally by program staff to manage case notes. The "type of service" field was used internally by the agency to track process outcomes and to audit case manager activity.

**Individual development plans.** Each month, case managers met with clients to review and update individual development plans. Each plan was comprised of personalized goals that were specific, measurable, attainable, realistic, and time-limited (SMART). These individual development plans were meant to facilitate the accomplishment of clients' goals within the domains of safety and security, physiological needs, and clients' personal aspirations related to employment, education, social support, and more. The goal forms were input into the Traffick911 online data management system by program staff after completing paper copies with clients in the field. The paper copies of the goal forms were available in both English and Spanish. The online system allowed program staff to specify that a goal was "met", "discontinued", or "in progress". An image of one goal form is included in S2 Fig; clients were permitted to set as many goals as they wished. The agency's target outcome was for at least 75% of clients to have developed at least one goal within 90 days of beginning services, and for Traffick911 clients to have met 2/3rds of their individual development plan goals during each quarter of the year.

**Readiness to change surveys.** Clients' readiness to change risky behaviors is a proxy for clients' readiness to exit sex trafficking, and it was assessed with a modified version of the 24-item University of Rhode Island Change Assessment (URICA) [34, 35]. The URICA is used to predict treatment outcomes and indicate progress during treatment. The items were modified such that "your problem" noted in each of the items was changed to "the game"–which is language commonly used by sex trafficking survivors to reference sex selling or trading behaviors. After member checking this language with Traffick911 program staff, several staff members requested that the research team make a modifiable version of the URICA available for cases in which "the game" could be psychologically triggering to clients. After numerous discussions with program staff, the research team helped the program staff develop a decision tree to guide the selection of a URICA scale for clients (Fig 2). For clients who could be triggered by "the game" language, program staff were able to use one high-risk behavior commonly associated with sex trafficking as a behavior targeted for changing. Examples included running away from home, sex while under the influence of drugs, sex with multiple partners, and more. Since this decision tree was developed collaboratively with program staff to meet

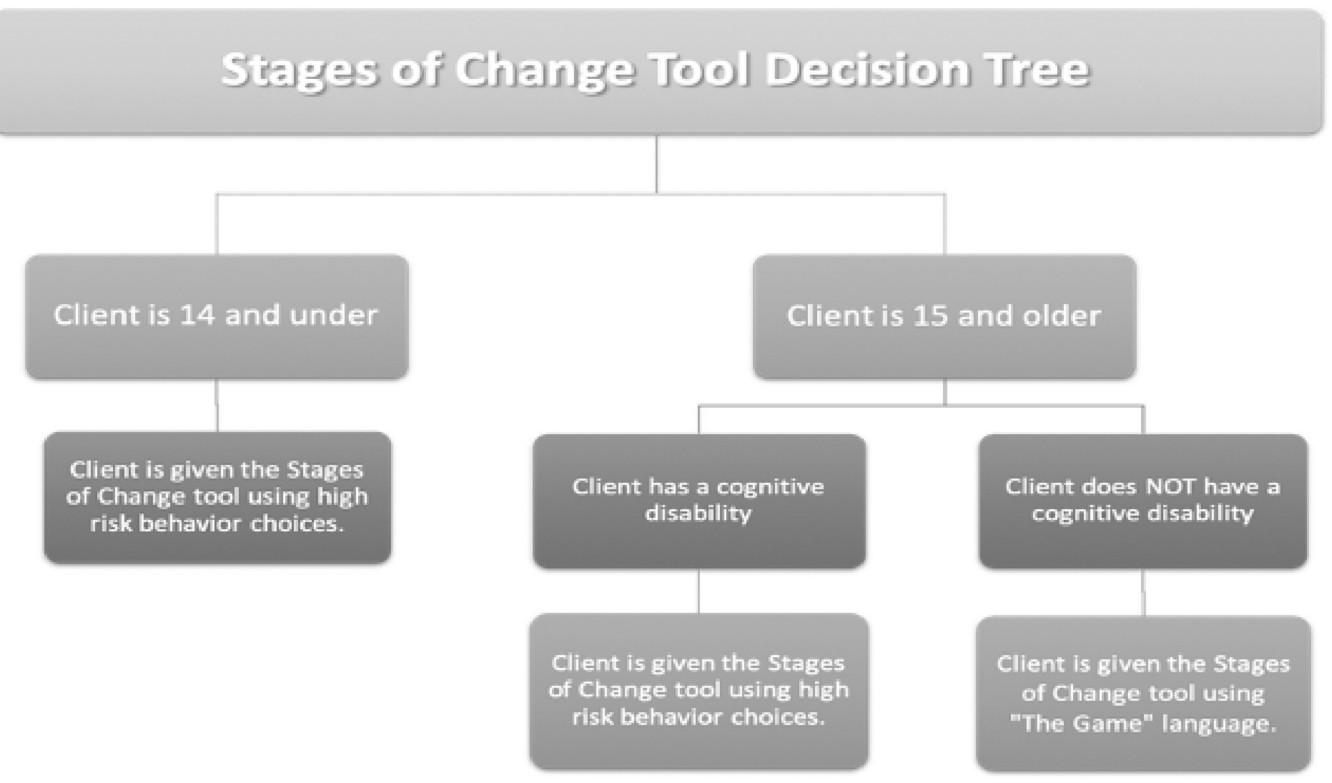

**Fig 2. URICA decision tree.** Agency's protocol to select appropriate stages of change survey.

program and client needs, researchers who adapt this protocol for future evaluations should make every effort to collaboratively adapt this sample decision tree to meet individual program needs. Further, it should be noted that the decision tree has its own limitations; the agency did not have a uniform procedure to assess cognitive ability in clients, and a case can be made that there are other factors besides age and cognitive ability that might necessitate using the "high-risk behavior" version of the tool. Researchers should approach the decision tree with some caution and be willing to adapt procedures to match agency and client needs. All versions of the URICA were available in both English and Spanish (See S1 File). Program staff completed the URICA with each client at baseline, six months, and twelve months, and input individual question data into Qualtrics for data management.

Using individual question data, the research team created a Readiness to Change score by summing items from each subscale, dividing by seven to get the mean for each subscale, and then summing the means from the Contemplation, Action, and Maintenance subscales and subtracting the Precontemplation mean (C + A + M–PC = Readiness). This is the Readiness score equation offered by the URICA's authors, and it has been used in other studies to investigate readiness to change substance use behaviors, smoking cessation, parenting behaviors, and more [36–38]. The Action and Maintenance subscale scores were used to measure treatment progress. The goal was for at least 50% of clients to progress in their stages of change score within one year of intake.

**Social support surveys.** Clients' social support was measured with the Multidimensional Scale of Perceived Social Support (MSPSS) [39]. The MSPSS is a brief research tool designed to measure perceptions of support from three sources: family, friends, and significant others. The scale is comprised of a total of 12 items, with four items for each subscale. The items were

averaged to obtain a total for each subscale and an overall score. The MSPSS was available in both English and Spanish (See S2 File). Program staff completed the MSPSS with each client at baseline, six months, and twelve months, and input individual question data into Qualtrics for data management. Traffick911's goal was for clients to demonstrate a 50% improvement on social support scores within one year of intake.

**Self-efficacy surveys.** Clients' self-efficacy was assessed with the Coping Self-Efficacy Scale, a 26-item measure of confidence coping with changeable and unchangeable life stressors [40]. Self-efficacy is an important component of Cimino's IEP model [23], and it is aligned with Traffick911's theory of change. Items refer to behaviors important to adaptive coping. Participants indicated their confidence performing each behavior on an 11-point scale (0 = "cannot do at all" to 10 = "certain can do"). An overall sum score was created, which can be used to measure change in coping skills. The Coping Self-Efficacy Scale was available in both English and Spanish (see S3 File). Program staff completed the Coping Self-Efficacy Scale with each client at baseline, six months, and twelve months, and input individual question data into Qualtrics for data management. The goal was for a 50% improvement on self-efficacy within one year of intake.

## Analyses

De-identified data was uploaded to SPSS for data cleaning and statistical analyses. Missing data was handled with maximum likelihood estimation or listwise deletion when there was more than 80% missing. To date, the research team has examined descriptive statistics (i.e., mean, standard deviation, medians, counts, percentage, etc.) to check for normality and assess assumptions for each variable, but data analysis remains ongoing at this time. Descriptive statistics, paired t-tests, and chi-square analyses will be used to determine whether the agency met its short- and long-term outcomes, as articulated in the Program Logic Model (Fig 1). Multivariate analyses like linear, logistic, and Poisson regressions will determine how doses of advocacy–as captured by the case note form–predict client outcomes as measured by the URICA, the MSPSS, and the Coping Self-Efficacy Scale, while controlling for a variety of demographic factors. This analysis will answer whether individual advocacy works to produce desirable client outcomes. Finally, a three-step latent profile analysis will be used to (a) estimate latent advocacy need profiles using case note data (e.g., high, medium, and low service contacts), (b) test for demographic differences in profile membership, and (c) test whether the profile membership predicts URICA readiness stages.

## Dissemination

The results of the evaluation will be disseminated in reports and white papers with Traffick911 and with the Office of the Texas Governor's Child Sex Trafficking Team as part of the PI's role as a member of the North Texas Academic Collaborative on Trafficking (NTACT)—an initiative launched by the state of Texas. Scholarly presentations and journal articles will be developed that focus on providing insight into the service outcomes for domestic minor sex trafficking survivors, which is an area of research that remains relatively unexplored [41].

## Conclusion

Youth sex trafficking is a significant concern in the North Texas region of the United States and beyond. This study protocol is designed to measure the mental and behavioral health outcomes of sex trafficked cisgender girls in the North Texas region who receive individual advocacy services from a hybrid community public health and social service program called Traffick911. Since research on the effectiveness of advocacy programs is lacking, this study

protocol may be adapted to other service contexts to measure whether and how similar programs decrease psychosocial distress and increase readiness to exit CSE.

## Supporting information

**S1 Fig. Case note form.** Screenshot of the case note interface form in Apricot.
(DOCX)

**S2 Fig. Individual development plan.** Form for goal development with clients.
(DOCX)

**S1 File. Readiness to change surveys.** URICA measure in English and Spanish.
(DOCX)

**S2 File. Social support surveys.** MSPSS measure in English and Spanish.
(DOCX)

**S3 File. Self-efficacy surveys.** Coping self-efficacy measure in English and Spanish.
(DOCX)

## Author Contributions

**Conceptualization:** Mary K. Twis, Andrea Cimino.

**Funding acquisition:** Mary K. Twis.

**Investigation:** Mary K. Twis.

**Methodology:** Mary K. Twis, Andrea Cimino.

**Project administration:** Mary K. Twis, Morgan Files.

**Supervision:** Mary K. Twis.

**Writing – original draft:** Mary K. Twis.

**Writing – review & editing:** Andrea Cimino, Morgan Files.

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
