## [Decision Letter · Decision Letter 0]

31 Jan 2022

PONE-D-21-27852

Does Individual Advocacy Work?: A Research and Evaluation Protocol for a Youth Anti-Sex Trafficking Program

PLOS ONE

Dear Dr. Twis,

Thank you for submitting your manuscript to PLOS ONE. After careful consideration, we feel that it has merit but does not fully meet PLOS ONE’s publication criteria as it currently stands. Therefore, we invite you to submit a revised version of the manuscript that addresses the points raised during the review process.

We look forward to receiving your revised manuscript.

Kind regards,

Johnson Chun-Sing Cheung, D.S.W.

Academic Editor

PLOS ONE

https://journals.plos.org/plosone/s/file?id=ba62/PLOSOne_formatting_sample_title_authors_affiliations.pdf”.

“This project was supported by a Victims of Crime Act (VOCA) federal pass-through grant awarded to the Office of the Texas Governor (OOG) Child Sex Trafficking Team and Traffick911. The sponsor of this protocol - Traffick911 - has assisted with the study design and data collection plan described in this protocol. Traffick911 administrators have approved the protocol described herein. The opinions, findings, conclusions, and recommendations expressed in this publication are those of the author(s) and do not necessarily reflect the views of the VOCA, OOG, or Traffick911.

https://www.traffick911.com”

**Comments to the Author**

Reviewer #1: This protocol describes a meaningful evaluation of an anti-sex trafficking program, and the resulting analyses will almost certainly be a meaningful addition to the field. A few revisions by the authors would greatly improve the clarity and accuracy of the protocol.

p2: The statistic for 300,000 at-risk youth should be clarified, as the bounds by which youth are considered at risk in this citation are not the same as those used later in the Methods to determine which individuals are “at risk” and eligible for the program. The text should be revised in these introductory paragraphs and in the Methods to make it clear to the reader that this discrepancy exists.

Abstract and throughout the main text: “Female youth” is used – Do the authors mean girls? Or will the program include all genders of individuals who were assigned female at birth (e.g., cisgender girls, transgender boys, non-binary youth, gender non-conforming youth, etc.)? This seems to be answered on page 8 when describing the inclusion criteria. However, there is a mismatch in the terms that should be corrected. “Female” youth should be changed to “cisgender girls” and “male” youth changed to “cisgender boys” throughout the text.

Additionally, as there is not a section for limitations of the protocol, the authors should make some mention (whether as part of the introductory paragraphs or in describing the inclusion criteria) of how the focus of this study on only cisgender girls may influence the scope of their findings and widen the systemic gaps (overlooking cisgender boys, transgender youth, non-binary youth, and gender non-conforming youth, etc.) already existing in the field studying minor sex trafficking/CSE. Though the authors make a reasonable case for this exclusion (p8), it should be clear how this decision will impact and limit the scope and applications of their work.

Figure 2 (and p12 text): It is not wholly clear from the in-text explanation why the two conditions in Figure 2 (age and disability status) are the decision points for whether “the game” is used as part of the URICA form. As the authors have noted that the language may need to shift away from “the game” for clients who would find it psychologically triggering, these conditions do not seem to capture all cases in which such language would be triggering or inappropriate. It would be helpful for the authors to explain the rationale for these conditions further.

Some clarification is needed throughout the text around the program goals, outcomes measured, methods, etc. for individuals in each of the following sub-populations receiving the program: (a) those who have experienced CSE previously but are not currently experiencing CSE (e.g., not in the last month or 6 months or some other time window); (b) those are currently experiencing CSE; and (c) those who are considered to be “at risk” for CSE based on explained criteria. A large portion of the instruments (e.g., Appendix 3) and the outcomes (e.g., URICA) and the organizational materials (e.g., Figure 1) seem to assume that clients will all be currently engaged in CSE/“the game” and prioritize exiting exploitation. However, these assumptions do not fit or apply to those who have a history of CSE but are not currently experiencing it or those who are just “at risk” of CSE. Additionally, the Abstract states that the sub-population of interest is just those “at-risk for sex trafficking” which seems to contradict with the main text. Thus, throughout the text (Abstract, main text, appendices), it would be helpful to have the authors clarify how the instruments will be applied to each sub-population and to note specifically how the central focus around “readiness to leave CSE” (stated in the Abstract) applies.

Reviewer #2: The article presents an evaluation protocol for evaluating the effectiveness of a youth anti-sex trafficking program, considering, whether female sex trafficked youth who receive individualized advocacy services experience changes in social support, self-efficacy, and readiness to change the behaviors that predict sex trafficking risk.

The proposal is quite interesting and necessary as it offers a protocol that can be adapted to other contexts and contribute to evaluate the effectiveness of other programs that intervene with young victims of sexual exploitation. Fortunately, there is an increasing amount of scientific literature on the causes of CSE, risk factors, victims' profiles, and consequences. However, there is little literature focusing on elements that favor the success and sustainable exit of sexually exploited victims over time and what elements contribute to it. Likewise, there are many intervention programs from different organizations and in different countries that help sex trafficking victims to cope with the consequences, have future opportunities and alternatives if they so wish. It is important that these programs have tools to help them evaluate their effectiveness, results, and impact, in order to improve or transfer their good practices. This article, with the protocol proposal can contribute to this purpose.

As a native Spanish speaker, I would like to suggest few recommendations regarding the Spanish translation:

a) Please consider, If possible, to check and validate the Spanish questions with CSE Spanish speaker survivors and be sure that they also use the term “la vida” as a synonym of “sex trafficking”. "The life" may work in the US with some Spanish-speaking population but it would not work in Latinamerica or Spain because victims do not colloquially refer to “la vida”. Maybe consider adding a definition of “la vida” with more synonyms in a footnote. Also, consider doing the same with the word “el juego” translated as “the game”. "El juego" can also be misunderstood. The word “el juego” in Spanish is associated with gambling. As a curiosity, on this website, you can consult the diversity of terms that exist in Spanish on this subject. https://www.curiosidario.es/prostitucion/

b) The words “circule el número” is not apropipate, it is an english translation from “circle” but “circule” in Spanish means “driving”, althoug of course “circule” in a questionnaire can be understood by spanish-speakers. My recommendation is to use the following expression “marque con un círculo la respuesta correcta….” “marque con un círculo el número que mejor describa su grado de acuerdo o desacuerdo con cada afirmación”. “Marque con un círculo…” is the standard and most common sentence used in Spanish questionioneirs and tests for "circle".

c) Regarding the sentence: “Pensé que había resuelto las razones por las que estoy en el juego, pero todavía sigo luchando con esas razones” consider substituting the word “razones” for “motivos” or “causas”. Also, saying “sigo luchando con esas razones” means “I keep fighting with those reasons” I believe authors prefer to say “against those reasons”. The correct full sentence in Spanish would be: “Pensé que había resuelto los motivos/ las causas por los que estoy en el juego pero todavía sigo luchando contra esos motivos/esas causas”. Also the sentence does not sound natural in Spanish, it is a literal translation. I consider it might be difficult to understand for a CSE survivor. I encourage the authors to validate these questions with Spanish CSE survivors.

d) Regarding the sentence “puede que tenga fallas”, please consider saying “puede que tenga fallos”. Fallas in femenine and plural is incorrect: https://dle.rae.es/falla Falla in singular means fault or defect. The correct is to say “fallos”, and more correctly the whole sentence should be “puede que cometa fallos” or “puedo tener defectos”. Also the word “participación” sounds very technical and strange in this sentence.

e) The word “seria” in the sentence “pensé que una vez que dejara el juego seria libre…” needs accent mark: “sería”. In this sentence is also hard to understand what is “el”, are you referring to the game? the subject is not clear. If you refer to “el” as the game, write it with accent mark: “él” https://www.rae.es/dpd/pronombres%20personales%20t%C3%B3nicos

f) Regarding the sentence “26. No actuar impulsivamente cuando esta bajo presión” the word “esta” needs accent mark “está”.

Most of the Spanish phrases sound strange with technical words that translated literally into Spanish are difficult to understand, do not sound natural, or may not even make sense. In addition to considering revising the Spanish and validating it with Spanish-speaking victims, it may be a good idea to add the recommendation or instructions that the questionnaire is answered with the help of professionals to ensure victims understand the questions correctly.

Reviewer #3: This paper presents a protocol to explore the effects of individualized advocacy services on social support, self-efficacy, and readiness to change among female youth sex trafficking victims in North Texas. The rationale for the protocol is clear, as there is limited evaluation data available on anti-sex trafficking advocacy programs. The protocol is well-written but would benefit from more detail in the areas outlined below. Providing greater context to these areas would greatly improve the protocol, especially when considering how the protocol could influence future evaluation efforts in this area of study.

• The first paragraph of the Methods section discusses how the Texas Governor’s Child Sex Trafficking Team was focused on creating Multi-Disciplinary Teams (MDT), yet there is no content on how Traffick911 relates to MDTs? Is Traffick911 an MDT? How does Traffick911 coordinate services across systems? Who are their partners when responding to reports of child sex trafficking? Since MDTs are referenced throughout the protocol, clarification is crucial.

• The logic model and associated written descriptions are incredibly illustrative and helpful to readers. What is missing is a discussion about why Traffick911? What about the organization positions them to administer this individual advocacy program? I can think of many reasons, including their central location, current service offerings, connections in the community, staff capacity, etc., but having these explicitly outlined in the paper would be helpful.

• Additionally, information about Traffick911’s current capacity for client direct services (i.e., how many women are reached by Traffick911) and how it relates to the proposed enrollment numbers would be important to include. Is there any data on the short-term outcomes outlined in the logic model? While available evaluation data may be limited for a number of reasons, including such information in the paper would provide readers with greater context about the reach/effectiveness of existing services.

• It also seems some of the outcomes in the logic model are listed specifically to address the goals of the proposed individual advocacy program (e.g., readiness to change, self-efficacy, and social support). Is the logic model presented for the individual advocacy program, or is this logic model adopted by Traffick911 as an organization? Initially, I thought it was the latter, as the goals of the individual advocacy program are at the individual level but would appreciate clarification on when this logic model was adopted and motivating factors.

• Given the study is happening during the COVID-19 pandemic, are there any implications that need to be included in the research protocol specific to this time period? Are there aspects of the protocol that would be unnecessary if not in a global pandemic?

• Does parental consent pose any challenges when providing services to at-risk youth, particularly those who run away from their home? What implications does this have for the study?

• When describing Individual Development Plans (and other measurement tools), there is mention the form is available in both English and Spanish; however, diversity of language was not described under the Study Site section. This information would be critical to assessing feasibility of implementing the program/measuring outcomes using existing tools.

• How have results from the University of Rhode Island Change Assessment (URICA) tool been reported in previous publications? Specifically, has the readiness equation been used before? The proposed equation seems appropriate; however, citations would strengthen this section.

• Traffick911’s theory of change is referenced in the Self-efficacy Surveys section; however, no citation/figure is included. The theory of change would be an excellent addition to the protocol, if available.

• In Figure 2, how will Traffick911 staff know if a client has a cognitive disability? Is this information obtained from partner agencies?

• The Analyses section would benefit from additional information on if/how confounding factors may be controlled for during regression analyses.

---

## [Author Response · Author response to Decision Letter 0]

2 May 2022

Please see my detailed response to the reviewers included on an uploaded Word Doc. We appreciate the reviewers' comments and have made every effort to incorporate their comments into the revised manuscript.

---

## [Decision Letter · Decision Letter 1]

6 Jun 2022

Does Individual Advocacy Work?: A Research and Evaluation Protocol for a Youth Anti-Sex Trafficking Program

PONE-D-21-27852R1

Dear Dr. Twis,

We’re pleased to inform you that your manuscript has been judged scientifically suitable for publication and will be formally accepted for publication once it meets all outstanding technical requirements.

Kind regards,

Johnson Chun-Sing Cheung, D.S.W.

Section Editor

PLOS ONE

---

## [Editor Report · Acceptance letter]

20 Jun 2022

PONE-D-21-27852R1 

Does Individual Advocacy Work?: A Research and Evaluation Protocol for a Youth Anti-Sex Trafficking Program 

Dear Dr. Twis:

I'm pleased to inform you that your manuscript has been deemed suitable for publication in PLOS ONE. Congratulations! Your manuscript is now with our production department. 

Kind regards, 

on behalf of

Dr. Johnson Chun-Sing Cheung 

Section Editor

PLOS ONE